# A Collaborative Perspective on Exploration in Reinforcement Learning

## Abstract

Exploration is one of the central topics in reinforcement learning (RL). Many existing approaches take a single agent perspective when tackling this problem. In this work, we view this problem from a different angle by taking a collaborative parallel-agent perspective. By doing this, we can not only learn with parallel agents, which is not fundamentally different by itself, but more importantly, it unlocks the possibility of introducing collaborative exploration and learning among these agents. We formulate this problem as *Collaborative Exploration* and propose concrete instantiations. We introduce a collaborative reward generator as a core component to induce collaboration, which can compute novelty of a state not only from one agent's own perspective, but also respect other agents' intrinsic motivation in pursuit of novelty. This leads to collaboration and specialization of each agent within the set of agents. In addition, we discussed how to effectively leverage the shared information from other agents in the data collection and evaluation phases, respectively. Experiments on the DMC benchmark tasks showcase the effectiveness of the proposed method. Code will be released: https://github.com/Anonymous/CE_URL.

## 1 Introduction

Reinforcement Learning (RL) (Sutton & Barto, 2018) has achieved great success across a wide array of applications. However, it typically requires a large number of interactions with the environment (Micheli et al., 2023; Kapturowski et al., 2023), which largely limits its practical application (D'Oro et al., 2023). While there are multiple reasons for the low sample efficiency, inefficient exploration is one of the major factors (Gao et al., 2022; Zhang et al., 2022; Hu et al., 2023). Therefore how to improve its sample efficiency by improving exploration is an important topic (Du et al., 2020; Pierrot et al., 2022; Zheng et al., 2022; Zhou & Garg, 2023).

A large body of work has been developed to address this issue from different aspects (Tiapkin et al., 2023; Zhang et al., 2023a). The common practice is to formulate the problem in the framework of a single agent RL problem, augmented with an additional intrinsic reward on top of the task reward (Badia et al., 2020; Lobel et al., 2023; Pathak et al., 2017; Burda et al., 2019; Seo et al., 2021; Jo et al., 2022; Yuan et al., 2023; Wang et al., 2023b). The intuition is that by the incorporation of this intrinsic reward, the policy can gain some learning signal even before encountering any useful task reward. This paradigm is very powerful and has led to many successful progresses in the past.

This work aligns with prior work in the direction of improving sample efficiency by designing better exploration strategy (Wang et al., 2023b; Yuan et al., 2023; Zahavy et al., 2023). The difference is that we explore a *collaborative* perspective (Du et al., 2023) in contrast to the commonly taken single agent perspective for exploration (Pathak et al., 2017; Burda et al., 2019; Seo et al., 2021). This is achieved by additionally introducing another dimension of multiple agents (Ding et al., 2023; Qiu et al., 2023) and taking their interactions into account during learning and exploration (Zhang et al., 2023c). The collaborative effects are achieved mainly from the following interconnected components:

1. **parallel agents**: instead of a single agent, we introduce a set of agents, each with its own learnable parameters and environment;

2. **collaboration and specialization**: during training, the knowledge among agents is shared by data and intrinsic reward. A *collaborative (intrinsic) reward generator* is used to encourage each agent not only guided by the intrinsic in the typical sense (e.g. novelty etc), but also to

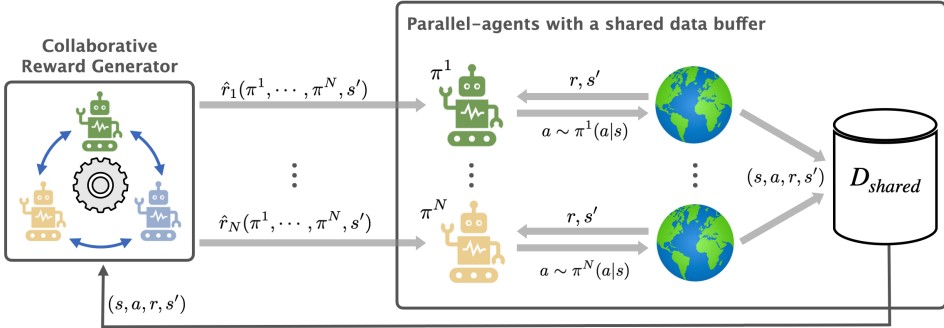

Figure 1: **The proposed Collaborative Exploration framework**: we achieve collaborative exploration via a collaborative reward generator $\mathcal{R}$ to share the novelty information among agents.

> be distinct from the other agents in the agent set, encouraging specialization of each agent; during unroll, each agent explores according to its specialization, while further taking other agent's unroll behavior into consideration.

It is important to emphasize that these components are inter-connected with each other and it is the interactions between them that make the full approach effective. For example, the parallel-agent structure, by itself, does not bring any essential differences, compared to the single-agent form. However, it unlocks the possibility of collaborative exploration and learning, once used together with the other components, as done in this work. Figure 1 shows an illustration of our framework.

The main contributions of this work are: (1) We present a collaborative perspective for improving exploration in RL. (2) We designed a contrastive collaborative exploration approach as an instantiation. (3) Experiments show that the proposed method outperforms other SOTA methods.

## 2 RELATED WORK

**Parallel Agents and Distributed RL.** From a model structure perspective, our method is similar to prior work on *Parallel Agents* and *Distributed RL* (Mnih et al., 2016; Espeholt et al., 2018; 2020; Petrenko et al., 2020; Mei et al., 2023). The goal of them is usually to achieve high data throughput (Liu et al., 2020) by adopting a large number of parallel agents and specialized architecture designs (Espeholt et al., 2020; Mei et al., 2023). Moreover, there is usually no explicit collaboration among the agents except for the shared replay buffer. Our work is different from this line of research in that we focus on the question of how to leverage explicit collaboration among different agents.

**MARL.** Multi-agent RL (MARL) aims to coordinate multiple agents to achieve a shared objective (Zhang et al., 2021). A typical setting of MARL is multiple agents residing in the same environment, where different agents have different roles to solve a cooperative/competitive task (Yu et al., 2022). These MARL agents share experiences and (or) weight parameters (Gupta et al., 2017; Terry et al., 2020; Christianos et al., 2021). In contrast to MARL, in our setting, each agent reside in its own environment and the agent's action will not influence the other agent's observations/rewards. Therefore, from the perspective of settings, ours is closer to that of the parallel agents than MARL.

**Concurrent RL.** In concurrent RL, multiple agents simultaneously interact with different instances of the same environment and share experiences with each other (Silver et al., 2013; Dimakopoulou et al., 2018; Taiga et al., 2022). A large body of work in this category studies some theoretical aspect of the problem including regret bounds etc. (Pazis & Parr, 2016; Dimakopoulou et al., 2018; Dimakopoulou & Van Roy, 2018; Taiga et al., 2022; Chen et al., 2022). Our work is aligned with this category in terms of settings, and focuses on improving exploration and sample efficiency and further investigates when and how to collaborate among the agents.

**Intrinsic Reward for Exploration.** One common form of intrinsic reward based exploration (Pathak et al., 2017; Burda et al., 2019; Seo et al., 2021; Jo et al., 2022; Yuan et al., 2023) is to modify the task MDP as $\mathcal{M}' = (\mathcal{S}, \mathcal{A}, \mathcal{P}, \hat{r}, \gamma, \mu)$, where $\hat{r}(s,a) = r_{\text{task}}(s,a) + \lambda r_{\text{intrinsic}}(s,a)$. $\lambda$ is a combination weight between the original task reward $r_{\text{task}}(s,a)$ and the intrinsic reward $r_{\text{intrinsic}}(s,a)$. While

the standard way under this context is to design $r_{\text{intrinsic}}(s, a)$ to reflect novelty etc., we leverage it with an additional purpose of encouraging specialization and collaboration in exploration. This is achieved by incorporating a collaborative reward generator in the learning process.

## 3  COLLABORATIVE EXPLORATION: FORMULATION

In this section, we introduce the main idea and formulation of the proposed method. Conceptually, the proposed approach first views the RL problem from the perspective of *multiple agents* (Wang et al., 2023a) and then introduces collaborations between them (Du et al., 2023). There are two key components in the current approach: parallel-agent formulation and collaborative exploration. In particular, the parallel-agent formulation is a necessary foundation for collaborative exploration.

### 3.1  LEARNING WITH A SET OF AGENTS

Common RL formulation typically adopts a single agent perspective. In standard off-policy RL algorithms such as SAC (Haarnoja et al., 2018), the single agent $A$ is composed of a single policy $\pi$ and a corresponding value function $Q$, represented as $A \triangleq (\pi, Q)$. In this work, we take a collaborative parallel-agent perspective instead by maintaining a set of agents $\mathbb{A} = \{A^i\}_{i=1}^N = \left\{(\pi^1, Q^1), \cdots, (\pi^i, Q^i), \cdots, (\pi^N, Q^N)\right\}$, where $N$ denotes the number of agents in the set. We denote the collection of policies as a policy set $\Pi = [\pi^1, \pi^2, \cdots, \pi^N]$ and the collection of value functions as a value set $\mathbb{Q} = [Q^1, Q^2, \cdots, Q^N]$. It's notable that this setting is also known as the Concurrent Exploration setting in some previous work (Silver et al., 2013; Pazis & Parr, 2016; Dimakopoulou et al., 2018; Dimakopoulou & Van Roy, 2018; Taiga et al., 2022; Chen et al., 2022).

During unrolling, each agent $A^i$ from the set interacts with its own environment according to its policy $\pi^i$. The data collected by all the agents are stored in a shared replay buffer $\mathcal{D}_{\text{shared}}$. The set of agents $\mathbb{A}$ are trained collectively on the jointly collected data. More specifically, each agent $A^i$ is trained not only on the data from its own unrolling, but also from the data collected by others (*c.f.* Fig. 1). Intuitively, this enables it to learn from more diverse data and share knowledge among agents.

### 3.2  COLLABORATIVE EXPLORATION AND LEARNING

The parallel-agent formulation presented above unlocks possibilities for different kinds of collaboration between agents. Actually, the shared replay buffer is the first form of collaboration via *data sharing*. To further leverage the presence of multiple agents in our formulation, we can incorporate other forms of collaboration. Here we show how to achieve collaborative exploration as an example.

**Collaborative Reward Generator.** To induce collaboration among the agents in $\mathbb{A}$, a collaborative reward generator $\mathcal{R}$ is introduced to generate unique intrinsic rewards (Jo et al., 2022) for each agent $A^i \triangleq (\pi^i, Q^i) \in \mathbb{A}$, with the awareness to others ($\{j, j \neq i\}$):

$$r_{\text{intrinsic}}^i(s, a) \triangleq \mathcal{R}(s, a, i, \{j, j \neq i\}). \tag{1}$$

Then each agent is trained under its own version of modified MDP as $\mathcal{M}^i = (\mathcal{S}, \mathcal{A}, \mathcal{P}, \hat{r}^i, \gamma, \mu)$ with its augmented reward $\hat{r}^i(s, a) = r_{\text{task}}(s, a) + \lambda r_{\text{intrinsic}}^i(s, a)$. $\lambda$ is a combination weight. More details on specific instantiations of $\mathcal{R}$ are presented in Section 4.1.

**Data Sharing and Relabeling.** To enhance the utilization of data through collaboration among agents and enable data sharing, the data collected by all the agents are stored in a shared replay buffer $\mathcal{D}_{\text{shared}}$ (Lee et al., 2021; Chen et al., 2022). For each agent during training, we first sample a batch from $\mathcal{D}_{\text{shared}}$, and then relabel the reward following Eqn.(1). Since Eqn.(1) is specialized to each agent while taking the rest of the agents into consideration, we can use the same batch of data sampled from the replay buffer to do specialized training for different agents.

**Collaborative Data Collection.** Since each agent is encouraged to be specialized in its own behavior, which can naturally generate more diverse behaviors and cover different spaces collectively during exploration and data collection (Sheikh et al., 2022; Parker-Holder et al., 2020). Furthermore, it is also possible to further incorporate certain forms of explicit knowledge-sharing mechanism between agents on which action should be taken as a form of collaborative data collection. We will explain in detail each component and example instantiations in the sequel.

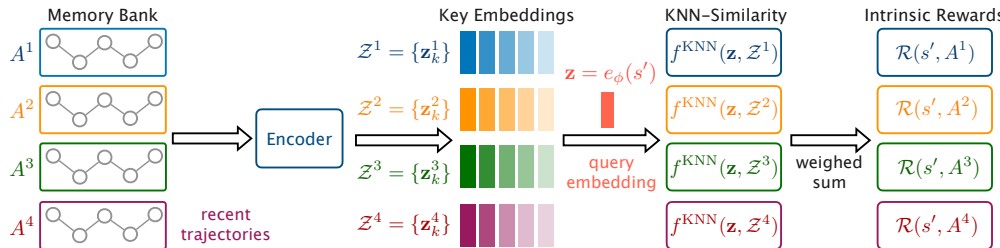

Figure 2: **Collaborative reward generator**: for a sampled afterstate $s'$, we compute its KNN cosine similarity w.r.t. each agent's recent trajectories and use the negative similarity as the intrinsic reward.

## 4 COLLABORATIVE EXPLORATION (CE)

To investigate the collaborative behaviors among agents in $\mathbb{A}$ and the impacts on exploration, we concentrate on two key questions: (1) How to collaborate? and (2) When to collaborate? For the first question, we mainly study the effectiveness of sharing novelty information among agents (Section 4.1). For the second question, we study the effects of incorporating collaborative behaviors at different phases: during the data collection, training and evaluation phases (Section 4.2), respectively.

### 4.1 HOW TO COLLABORATE

We first introduce how to collaborate in $\mathbb{A}$ by sharing the novelty information with a *collaborative reward generator* $\mathcal{R}(s, a, i, \{j, j \neq i\})$. In a nutshell, the intrinsic reward (Burda et al., 2019; Jo et al., 2022; Wang et al., 2023b; Yuan et al., 2023) for each agent is calculated collaboratively, not only considering its own knowledge, but also taking other agents into consideration (Peng et al., 2020). Here, we introduce an instantiation which uses a contrastive encoder (Ermolov et al., 2021).

**Collaborative Contrastive Reward.** The main idea of the collaborative intrinsic reward is that given any transitions, to calculate intrinsic reward for a particular agent, we consider not only the behaviors of this agent, but the behaviors of all other agents as well. Specifically, to calculate the intrinsic reward for a transition $(s, a, r, s')$ sampled from the shared replay buffer, we first use a learnable encoder $e_\phi(\cdot)$ to extract a feature vector (referred to as query embedding) from the afterstates $s'$ as $\mathbf{z} = e_\phi(s')$. As a practical way to approximately capture the agent behaviors using data, we maintain a memory bank, which stores recent trajectories for each agent.

To measure the similarities of an afterstates $s'$ w.r.t. a particular agent $A^i$, we calculate the similarity of the query embedding $\mathbf{z}$ w.r.t. the embeddings of the states in the memory bank for $A^i$ ($\mathcal{Z}^i \triangleq \{\mathbf{z}_k^i\}$, referred to as key embeddings), and take the negative similarity as intrinsic reward w.r.t. $A^i$:

$$r(s', A^i) = -f^{\mathrm{KNN}}(\mathbf{z}, \mathcal{Z}^i), \tag{2}$$

where $f$ and $f^{\mathrm{KNN}}$ denote the cosine similarity and KNN cosine similarity respectively. With $r(s', A^i)$ only, the agent is encouraged to visit states that are novel from its own perspective (*ego*). To further induce collaborations among agents, we introduce inter-agent terms (*mutual*) and use the following collaborative reward for the agent $A^i$:

$$\mathcal{R}(s', A^i) = \underbrace{w * r(s', A^i)}_{\text{ego}} + \underbrace{(1 - w) * g\big(\{r(s', A^j)\}_{j \neq i}\big)}_{\text{mutual}}, \tag{3}$$

where $g(\cdot)$ is an aggregation operator such as mean/max/min and $w$ is a weight parameter. In the experiment, we use mean aggregator and $w = 0.5$. This process is illustrated in Figure 2. Intuitively, Eqn.(3) encourages the agent to visit states that are not only novel from its own perspective, but also to respect other agents' intrinsic motivation in pursuit of novelty. Alternatively, a strong motivation of one agent in pursuit of a state will reduce the motivations of others on the same state.

We train the encoder $e_\phi(\cdot)$ using the InfoNCE loss with a temperature parameter $\tau$:

$$\mathcal{L}_{\text{encoder}} = -\mathbb{E}\left[\log \frac{\exp(f(\mathbf{z}_i, \mathbf{z}_j)/\tau)}{\sum_{k, k \neq i} \exp(f(\mathbf{z}_i, \mathbf{z}_k)/\tau)}\right] \tag{4}$$

We select nearby samples (Stooke et al., 2021) as the positive pairs $(\mathbf{z}_i, \mathbf{z}_j)$, *i.e.*, $s_i$ and $s_j$ are within 5 timesteps, and use the other positive samples in a batch as the negative pairs $(\mathbf{z}_i, \mathbf{z}_k)$.

## 4.2 WHEN TO COLLABORATE

In Section 4.1, we have incorporated collaboration during *training*. There are also other scenarios where it is possible to leverage collaboration. In this subsection, we explore additional scenarios for applying collaborative information among agents beyond what was discussed above.

**Collaborative Data Collection.** Besides collaborating by sharing the novelty information, we can also collaborate in $\mathbb{A}$ by sharing the policy information. In this subsection, we introduced a simple collaborative exploration strategy by leveraging multiple policies for joint exploration. Given an agent set $\mathbb{A}$ and the policy set $\Pi = [\pi^1, \cdots, \pi^N]$, the $i$-th agent receives the observation $s_t$ from its environment at timestep $t$. We then use each agent to sample $M$ actions, *i.e.*, the $j$-th agent samples $\{a_1^j, \cdots, a_M^j\}$. Similar to the $\epsilon$-greedy exploration (Sutton & Barto, 2018), instead of sampling $\pi^i(s_t)$ directly, we select the following action for the $i$-th agent with probability $\epsilon$:

$$a^i = \max_k \sum_{j \neq i}^{N} \sum_{h=1}^{M} \|a_k^i - a_h^j\|^2. \tag{5}$$

The intuition of the proposed $\epsilon$-collaborative exploration is to let the agent select actions that are less similar to the other agents in order to explore more diverse behaviors.

**Collaborative Evaluation.** After learning a set of agents $\mathbb{A}$, we face another question of how to select actions in the evaluation phase. This might not necessarily be a problem when all agents are comparably good, *i.e.*, in some simple tasks. However, randomly picking one agent from $\mathbb{A}$ for the evaluation could lead to unexpected bad performance in complex environments under some edge cases. To mitigate this issue, we propose to learn an extra classifier $c_\psi(s)$ to select which agent to take action in the evaluation. Simply, we can use the softmax function $c_\psi(s) = \text{softmax}(Q^i(s, \pi^i(s)))$ as the classifier if all agents use the same hyper-parameters and $Q(s, a)$ functions are in the same scale. Alternatively, we can learn another value function $c_\psi(s) = \arg\max_a Q_\psi^e(s, a)$ where the action $a$ is the agent index. For example, $Q_\psi^e(s, i)$ is the cumulative reward starting from $s$ following policy $\pi^i$.

## 5 EXPERIMENT

In this section, we mainly focus on the following questions: (1) How does the proposed method compare against other baselines? (2) Can we generalize the idea of collaborative reward to other existing methods? (3) Does collaborative exploration really help to generate more diverse behaviors? (4) Are the collaborative information useful in the data collection and evaluation phases?

### 5.1 EXPERIMENT SETTING

**Tasks and Settings.** We use 15 tasks from the DeepMind Control Suite benchmark (referred to as DMC15) (Tassa et al., 2018) in the experiments. We use SAC (Haarnoja et al., 2018) as the backbone algorithm. The size of $\mathbb{A}$ is set to 4 and we run 1e6 environment steps on each task, where each agent runs for 2.5e5 environment steps. We do 4 gradient updates per environment step to match the total gradient updates for each agent as in baselines. The training is repeated with 5 random seeds. In the following experiments, we always select the first agent for evaluation. Apart from the results presented in the sequel, additional results are deferred to Appendix D due to space consideration.

**Baselines.** We compare CE to the following baselines: (1) SAC (Haarnoja et al., 2018): the basic single agent baseline, which is also used as the backbone algorithm in CE; (2) Replica: parallel SAC agents with a shared buffer and each agent has its own environment; (3) A2C: a basic distributed RL algorithm; (4) DIAYN (Eysenbach et al., 2018): a two-stage method to learn skill-conditioned policy; (5) RND (Burda et al., 2019): using the prediction error w.r.t. a random target network as intrinsic reward; (6) DiCE (Peng et al., 2020): collaborative exploration using a diversity regularization; (7) SUNRISE (Lee et al., 2021): reweighing samples according to uncertainty in ensemble RL.

### 5.2 PERFORMANCE ON THE DMC15 BENCHMARK TASKS

We first compare the proposed method (CE), to other baselines with state-based inputs. Table 1 reports the mean and standard deviation results on the DMC15 benchmark. We also report the results of the interquartile mean (IQM) performance aggregated over all the tasks following (Agarwal et al., 2021), as shown in Figure 5.2. The running time of CE is around 2 times of training a single agent.

Table 1: **Average return on DMC15**: CE generally outperforms other baselines.

| Environment | Single Agent | Parallel Agents | | Intrinsic Reward | | Collaborative Agents | | |
| --- | --- | --- | --- | --- | --- | --- | --- | --- |
| | SAC | Replica | A2C | DIAYN | RND | DiCE | SUNRISE | CE |
| acrobot-swingup | 9.9 (4.8) | 22.8 (17.6) | 39.2 (4.7) | 22.3 (17.4) | 11.3 (5.9) | 15.2 (11.8) | 5.7 (1.3) | **54.6 (30.3)** |
| cheetah-run | 847.3 (17.9) | 869.2 (19.8) | 271.4 (58.5) | 820.2 (39.2) | 857.4 (25.1) | 868.8 (8.7) | 810.9 (28.5) | **877.8 (4.1)** |
| hopper-hop | 148.2 (49.5) | 164.6 (93.3) | 20.4 (26.3) | 182.8 (55.0) | 164.7 (34.4) | 167.5 (62.6) | 188.0 (78.1) | **306.8 (20.6)** |
| hopper-stand | 784.7 (242.9) | 825.2 (205.5) | 69.7 (79.5) | 638.3 (237.9) | 809.7 (122.1) | 581.6 (274.3) | 733.0 (196.3) | **933.3 (7.8)** |
| humanoid-run | 147.5 (15.8) | 134.2 (27.9) | 0.9 (0.1) | 125.8 (23.6) | 110.6 (55.0) | 141.8 (24.0) | 136.4 (33.4) | **167.3 (6.4)** |
| humanoid-stand | 429.6 (243.1) | **844.2 (12.4)** | 5.3 (0.2) | 660.3 (156.9) | 744.8 (113.1) | 662.2 (109.3) | 732.8 (82.9) | 836.4 (37.4) |
| humanoid-walk | 494.1 (41.3) | 561.7 (32.7) | 1.5 (0.1) | 521.0 (48.6) | 463.2 (19.1) | 481.3 (42.3) | 538.3 (20.6) | **574.6 (20.2)** |
| finger-turn_hard | 826.8 (71.8) | 841.8 (47.5) | 204.6 (82.6) | 809.1 (18.7) | 763.1 (87.1) | 858.4 (66.5) | 812.0 (57.3) | **882.6 (21.1)** |
| pendulum-swingup | 652.1 (298.6) | 345.0 (320.8) | 50.3 (2.2) | 623.5 (284.8) | 836.0 (10.1) | 831.1 (12.2) | 688.3 (115.8) | **841.4 (4.3)** |
| quadruped-run | 838.9 (53.6) | 853.4 (54.6) | 185.9 (75.0) | 851.9 (56.0) | 837.9 (50.3) | 881.4 (18.4) | 841.8 (35.7) | **895.3 (25.3)** |
| quadruped-walk | 940.0 (21.1) | 945.6 (7.2) | 128.1 (45.1) | 919.9 (27.3) | 806.5 (200.1) | **951.4 (5.1)** | 835.6 (195.4) | 942.3 (22.3) |
| reacher-hard | 948.0 (8.4) | 963.4 (15.6) | 829.5 (76.0) | 957.9 (19.2) | 954.8 (9.6) | 962.2 (17.5) | **969.7 (2.5)** | 956.3 (13.6) |
| walker-run | 711.2 (59.5) | 767.7 (21.5) | 305.7 (162.8) | 729.0 (20.5) | 759.1 (26.3) | 795.6 (3.1) | 699.0 (49.8) | **799.7 (5.0)** |
| fish-swim | 205.8 (31.9) | 152.4 (30.6) | 84.8 (8.9) | 304.2 (116.4) | 260.2 (69.2) | 139.1 (38.6) | 196.5 (67.5) | **305.2 (15.4)** |
| swimmer-swimmer6 | 282.2 (39.5) | 288.5 (44.6) | 240.0 (20.6) | **332.5 (40.4)** | 300.4 (36.7) | 280.0 (50.4) | 290.4 (45.9) | 298.6 (47.8) |

We can observe that CE generally outperforms or matches the performance of other baselines in most tasks. The performance gap between Replica and SAC indicates that joint learning with a set of different policies can help to improve the exploration. On the other hand, the performances of RND are usually worse than CE, which proves the advantage of sharing data among agents. Further, CE outperforms DiCE by a large margin, which showcases the benefits of sharing novelty information among agents. A2C usually underperforms other baselines, where the collected samples are discarded once used. The performance gap between DIAYN and CE shows the difficulty in identifying the correct skills in the selected tasks. The comparison between CE and SUNRISE indicates that the proposed collaborative reward generator can achieve collaborative exploration more effectively. These results show that the proposed multiple agents setting with a collaborative reward generator can help to achieve more efficient exploration and better performance.

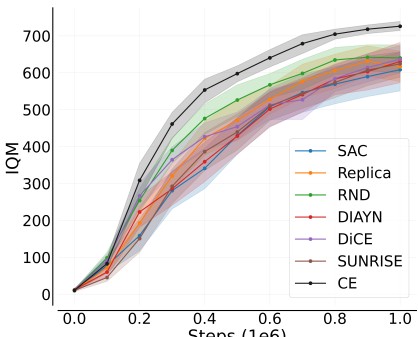

Figure 3: **The overall IQM score:** the proposed CE method achieves a higher IQM score on the DMC15 benchmark.

## 5.3 PERFORMANCE ON VISUAL TASKS

We also validate the effectiveness of the proposed method with complex visual inputs as shown in Figure 5.3. For the pixel-based tasks, we use the DrQ (Laskin et al., 2020; Yarats et al., 2021) as the backbone algorithm and use an agent set of size 2. We compare CE to DrQ, a Replica version of DrQ, RND and SUNRISE. We can observe that CE achieved the best performance in all four tasks, which suggests that our method is effective for both state-based and pixel-based inputs.

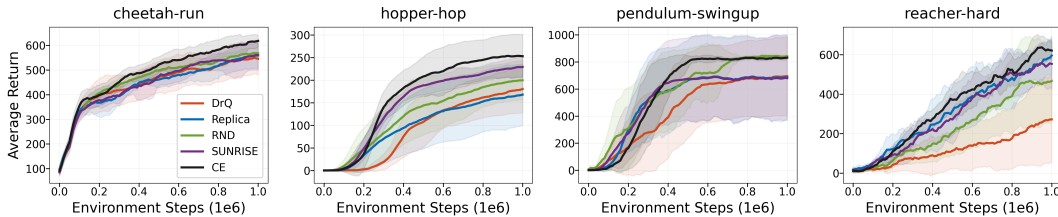

Figure 4: **Average return on pixel-based tasks**: CE also outperforms baselines with image inputs.

## 5.4 GENERALIZE THE IDEA OF COLLABORATIVE REWARD TO OTHER METHODS

Actually, the proposed collaborative reward is a general idea that can be implemented using different existing curiosity-based exploration methods (Pathak et al., 2017; Burda et al., 2019; Seo et al., 2021). Here, we introduce another variant using the standard RND module. Simply, instead of learning a single RND module for each parallel agent with its own data, we can learn the RND module using the shared buffer, as shown in Figure 14. By training the RND module on the shared buffer, the

Table 2: **CE is generally applicable:** combining the collaborative reward with RND is also effective.

| Methods | acrobot-swingup | cheetah-run | hopper-hop | hopper-stand | humanoid-stand | humanoid-walk | reacher-hard |
|---|---|---|---|---|---|---|---|
| RND | 11.3 (5.9) | 857.4 (25.1) | 164.7 (34.4) | 809.7 (122.1) | 744.8 (113.1) | 463.2 (19.1) | 954.8 (9.6) |
| + Replica | 24.7 (14.9) | **882.5 (13.1)** | 227.5 (28.1) | 925.4 (14.7) | 625.4 (228.0) | 522.8 (31.5) | 953.1 (26.7) |
| + CE (ours) | **34.5 (24.3)** | **884.6 (10.6)** | **267.9 (53.5)** | **935.3 (8.3)** | **809.9 (47.9)** | **558.6 (30.4)** | **962.1 (8.0)** |

computed intrinsic reward naturally contains the curiosity information w.r.t. the holistic agent set, hence leading to a collaborative exploration behavior to visit the states that are less visited by all agents. In addition, we can extend this baseline variant to learn a separate RND module for each agent on the shared buffer to increase diversity.

To validate the effectiveness of applying collaborative reward with RND, we compare the performance of different RND variants in Table 2. The RND corresponds to a single RND agent, RND+Replica learns a separate RND module using each agent's own data as in Figure 14(a), and RND+CE denotes the approach that applies the idea of CE to RND (more details are deferred to Section D.8) . From Table 2, we can observe that simply applying Replica to RND has some improvements over vanilla RND on some tasks but not significant overall. Applying CE to RND can effectively improve the performance over both RND and RND+Replica.

### 5.5 DOES COLLABORATIVE EXPLORATION HELP TO GENERATE MORE DIVERSE BEHAVIORS?

We first conduct a case study on a maze task, where an RL agent starts from the left and aims to arrive at either of two goal positions. In Figure 5, the top row shows the results of two non-collaborative agents, where each agent is trained individually with its own data. The bottom row shows the results of two collaboratively trained agents (CE). To visualize the behavior of each individual agent, we colored one agent's trajectories in blue and the other agent's trajectories in brown for both methods. We plot one trajectory per 10 trajectories for each agent, and the four columns correspond to the results when each agent runs for 100/200/300/400 trajectories. We can observe that the two collaborative agents are able to collect more diverse trajectories, while the two parallel agents fail to find the second goal position. This simple toy example illustrates that by using the idea of collaborative intrinsic reward, agents in the agent set can achieve more efficient exploration from a holistic view.

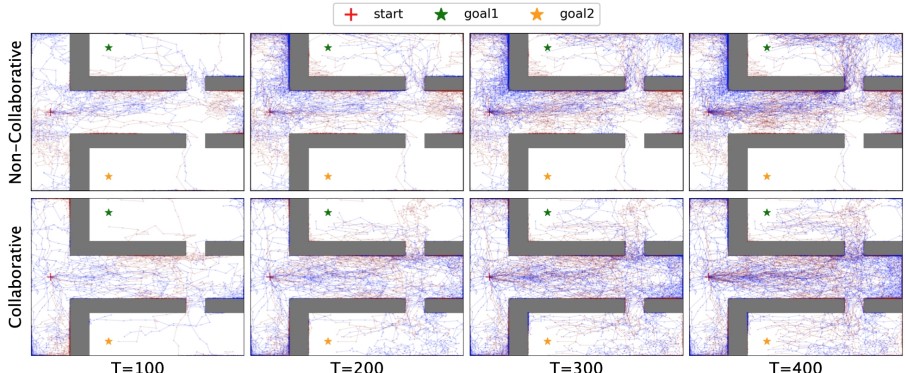

Figure 5: **Evolution of trajectories on the Maze task** at different training steps along the training process. (Top) non-collaboratively trained agents (Bottom) collaboratively trained agents.

We conduct a second case study on the MuJoCo (Todorov et al., 2012) environments, where we use the KL divergence $D_{KL}(\pi_i, \pi_j)$ and mutual information $I(S; A)$ to quantitatively approximate the diversity between different policies. To compute these metrics, we first use saved checkpoints to collect some trajectories $\{s_1, s_2, \cdots, s_T\}$. We then use the agents from $\mathbb{A}$ to sample actions for each collected state and measure the KL divergence $D_{KL}(\pi_i(a|s), \pi_j(a|s))$ between the $i$-th agent's policy and the $j$-th agent's policy. Next, we use the MINE lower bound (Belghazi et al., 2018) to estimate the mutual information $I(S; A)$ between the collected states and actions. Here, a smaller $I(S; A)$ means a higher uncertainty in action $a$ given state $s$ (Ma et al., 2023). Therefore, a larger $D_{KL}$ and a smaller $I(S; A)$ refer to more diverse policies. In the experiment, we use saved checkpoints to collect 3000 trajectories to compute the $D_{KL}$ and $I(S; A)$ metric for CE and Replica SAC agents. We report the average metrics and average return across 5 random seeds in Table 3. We

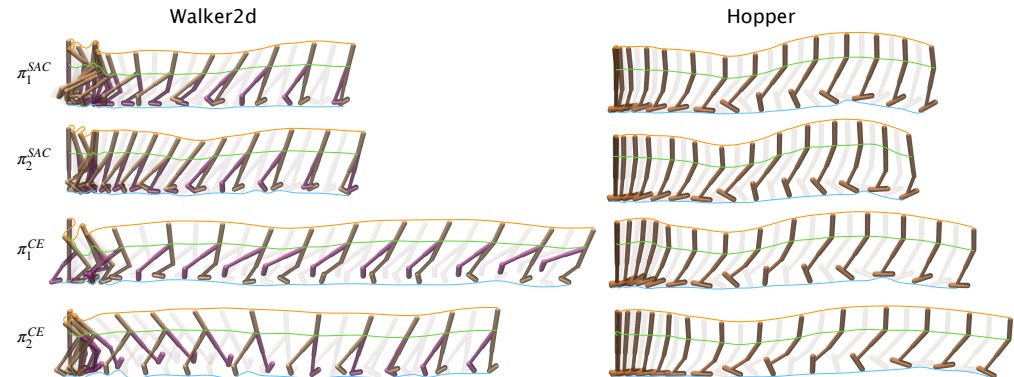

Figure 6: **Trajectories in the Walker2d and Hopper tasks**: The trajectories for some key points, *i.e.*, head, waist and foot, are colored in orange, green and blue. We can observe that the CE agents (last two rows) display more diverse gaits than the SAC agents (first two rows).

Table 3: **Diversity of agents:** we use mutual information and KL divergence as diversity proxies.

|  | Ant-V3 | | | HalfCheetah-V3 | | | Hopper-v3 | | | Walker2d-V3 | | |
|---|---|---|---|---|---|---|---|---|---|---|---|---|
| Metrics | $D_{KL}$ | $I(S;A)$ | $R$ | $D_{KL}$ | $I(S;A)$ | $R$ | $D_{KL}$ | $I(S;A)$ | $R$ | $D_{KL}$ | $I(S;A)$ | $R$ |
| Replica | 22.754 | 2.988 | 4190.5 | 1.912 | **3.068** | **11552.8** | 4.044 | 1.490 | 2674.1 | 9.191 | 2.771 | 4171.1 |
| CE | **23.611** | **2.751** | **4613.5** | **2.674** | 3.095 | 11545.1 | **5.488** | **1.444** | **3325.6** | **13.157** | **2.587** | **4876.6** |

can observe that the CE agents usually have larger $D_{KL}$ and smaller $I(S;A)$ metrics, except for the HalfCheetah-V3 task, where both CE and Replica SAC agents learned near-optimal policies.

We further use the checkpoints at 2e5 steps to plot trajectories (Figure 6) for each agent in the Walker2d-v3 (left column) and Hopper-v3 (right column) tasks (Janner et al., 2022). We can observe that the CE agents (last two rows) display more diverse gaits than the Replica SAC agents (first two rows), which is in accordance with the results in the Table 3.

## 5.6 IS COLLABORATIVE DATA COLLECTION USEFUL?

In this subsection, we investigate the usage of collaborative information during data collection. We compare the following baselines: (1) Replica: each agent independently interacts with its own environment; (2) $\epsilon$-greedy: each agent has probability $\epsilon$ to select a random action; (3) UCB: a baseline from the SUNRISE (Lee et al., 2021) that selects action w.r.t. the uncertainty of the learned $Q$-functions, i.e., $a_t = \max_a\{Q_{\mathrm{mean}}(s_t, a) + \lambda Q_{\mathrm{std}}(s_t, a)\}$; (4): Softmax: constructing a mixture policy using the softmax function w.r.t. each agent's $Q$-function; (5) $\epsilon$-collaborative: selecting action that maximizes the differences w.r.t. other agents' sampled actions as described in the Eqn.(5).

In the experiment, we use $\lambda = 1$ for the UCB method, $\epsilon = 0.2$ and $M = 10$ for the $\epsilon$-collaborative method. $\epsilon$-greedy method also uses $\epsilon = 0.2$. From the Figure 7, we can observe that simply selecting the most dissimilar actions outperforms the naïve Replica baseline in three out of four tasks, while the UCB and Softmax method only outperform the baseline in one task. The main reason is that UCB and Softmax exploration methods depend on the learned $Q(s, a)$ functions, which are usually inaccurate at the early stage of training and prevent the agent from collecting useful samples. These results show the efficacy of the proposed $\epsilon$-collaborative exploration strategy for collaborative data collection. Moreover, $\epsilon$-collaborative only adopts the policy information, so we can use different hyper-parameters, *i.e.*, discount factor $\gamma$, for each individual agent in the $\mathbb{A}$ for more diversities.

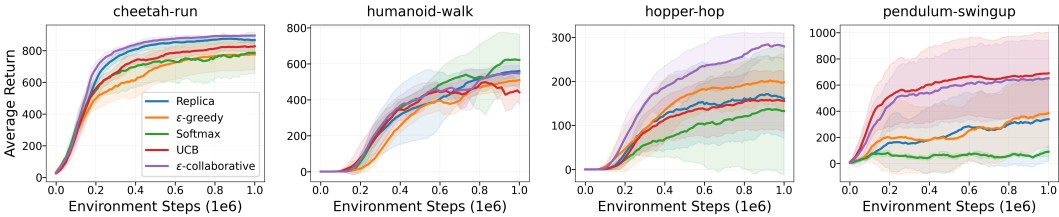

Figure 7: **Collaborative data collection:** a comparison of different unroll methods.

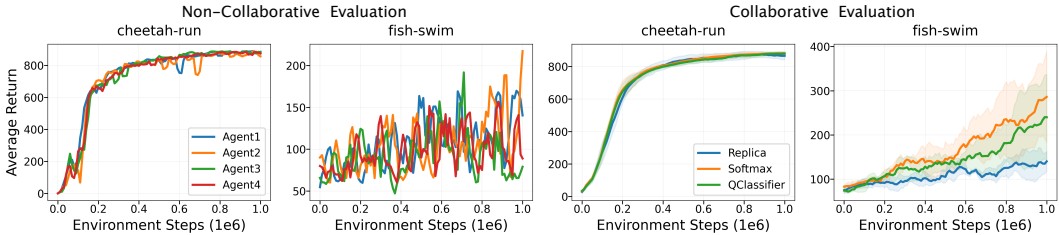

Figure 8: **Collaborative evaluation could help to mitigate the high variance issue:** (Left) Agent performance could vary depending on the task. (Right) Applying collaborative evaluation helps to mitigate the performance variance in some complex tasks, *i.e.*, fish-swim.

### 5.7 IS COLLABORATIVE EVALUATION USEFUL?

In this subsection, we study the efficacy of using collaborative information during the evaluation. We compare the following three methods: (1) Replica: we always by default select the first agent in the evaluation; (2) Softmax: we select actions according to a softmax policy $\text{softmax}(Q_i(s, \pi^i(s)))$ as introduced in the PEX (Zhang et al., 2023b); and (3) QClassifier: we learn an extra value function $Q_\psi^e(s, a)$ where agent index is the action. In the experiment, we simply use the multi-step cumulative reward as the Monte Carlo target and learn $Q_\psi^e(s, a)$ as in a regression problem. We set the multi-step horizon to be 100 and use $Q_\psi^e(s, a)$ to select the agent for evaluation for the next 100 steps. In Figure 8, we first plot the evaluation curves for the 4 Replica agents in the cheetah-run and fish-swim tasks for one random seed. Compared with the simple cheetah-run task, the agent performances have much higher variances in the fish-swim task. We then plot the evaluation curves for the two proposed collaborative evaluation methods across 5 random seeds. We can observe that the proposed collaborative evaluation methods could effectively reduce the variance and improve the performance.

### 5.8 EXPERIMENTS WITH MULTIHEAD CE

In this subsection, we further compare to a multihead variant of CE, where each agent shares the same torso network and uses different heads as the policy/value function. As we can observe from Figure 9, the multihead variant generally performs slightly worse than CE. One main reason is that the agent policies are less diverse in multihead CE when we use a shared torso network for each agent. On the other hand, the advantage of using the multihead CE is that it reduces the computation complexity by sharing parameters among agents. The superior performance of multihead CE against Replica agent again demonstrates the benefits of leveraging collaborative exploration.

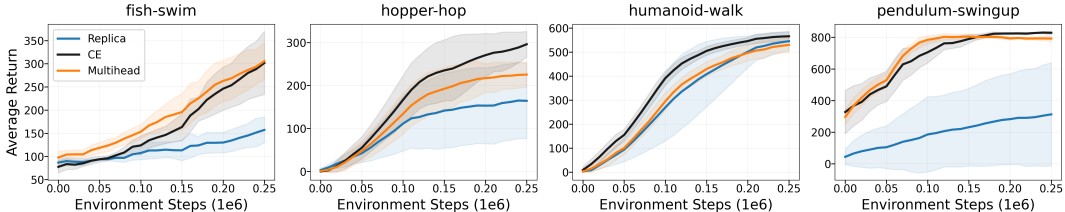

Figure 9: **Multihead CE:** a multihead variant where each agent shares the same torso network.

## 6 CONCLUSION

In this work, we focus on the problem of improving exploration in RL. Different from commonly used single-agent formulations, we take a parallel-agent perspective and use a collaborative reward generator to introduce collaborations between the agents, by sharing information among them when computing the intrinsic reward. Then we further investigated the questions of how and when to collaborate in the agent set to learn more diverse behaviors. We discuss the usage of collaboration in the data collection and evaluation phases beyond training. In addition to the proposed instantiation, the idea of collaborative exploration can also be integrated with existing intrinsic reward approaches easily. Experiments on different benchmark tasks demonstrate the efficacy of the proposed method.

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

APPENDIX

In the appendix, we first introduce some related work and discuss the connections and differences of our work with prior work. Later, we discuss the limitations and future directions of this work. Lastly, we report the details of experiment setups and additional experiment results.

## A    EXTENDED BACKGROUND MATERIALS

In this section, we discuss the connections and differences of our work with prior work.

**Multi-Agent RL**. Our problem setting with an agent set is very closely related to the *Multi-agent RL* (MARL) (Zhang et al., 2021; Jin et al., 2022; Qiu et al., 2023; Samvelyan et al., 2023) setting. In MARL, different agents operate in the same environment and share the state and reward information (Li et al., 2022), while in our setting each agent independently interacts with its own environment. Moreover, agents in the MARL setting usually work together to accomplish the same task, *i.e*, defeating enemies in the Dota2 game (Zhang et al., 2020a; Castanyer, 2023). However, the agent in our setting always focuses on its own task and its action will not affect the other agents. The proposed multihead variant of CE is also related to the parameter sharing based MARL (Gupta et al., 2017; Terry et al., 2020; Christianos et al., 2021; Yu et al., 2022), where the agents share some or all parameters. Parameter-sharing helps to reduce the number of trainable parameters and scale to more agents (Christianos et al., 2021). Parameter-sharing is usually used in MARL to learn cooperative policies (Gupta et al., 2017; Yu et al., 2022), while the original proposed CE is inclined to introduce more policy diversities via different policy parameters.

**Collaborative RL**. Our work is also closely related to some prior work in the literature on the Collaborative RL (Lin et al., 2017). For example, CERL (Khadka et al., 2019) employs a gradient-free neuroevolution method to learn a set of agents with different time horizons. CMAE introduced a shared common goal to achieve cooperative exploration in the MARL setting (Liu et al., 2021). Role Diversity provided theoretical analysis and evaluation metrics to measure a cooperative MARL task (Hu et al., 2022). InfoMARL (Nayak et al., 2023) aggregates the information of neighborhood of agents via a Graph Neural Network for the multi-agent navigation tasks.

**Distributed RL**. From a model structure perspective, our method is similar to prior work on the *Distributed RL* (Mnih et al., 2016; Espeholt et al., 2018; 2020; Petrenko et al., 2020; Mei et al., 2023), where multiple agents are interacting with their own environments simultaneously. Distributed RL usually adopts a large number of parallel agents to accelerate the exploration (Espeholt et al., 2020). Distributed RL focuses on improving the data throughput (Liu et al., 2020) via specialized infrastructures or architecture designs (Mei et al., 2023). However, we use a much smaller agent number than the Distributed RL and we only focus on leveraging the collaborative information from different agents to achieve the collaborative exploration. One common challenge of our method and the Distributed RL is the off-policy samples (Espeholt et al., 2018). We simply address this issue by using the gradient clipping training technique, and leave a combination of more advanced off-policy evaluation methods with our method for future work.

**RL with Policy Set and Ensemble RL**. Our work is related to work learning a set of policies (Badia et al., 2020; Lee et al., 2021; Zhang et al., 2023b; Sheikh et al., 2022). NGU (Badia et al., 2020) learns a set of policies each associated with a different discount factor. SUNRISE (Lee et al., 2021) maintains an ensemble of policies and selects an action from their proposals for exploration using UCB-base criteria. PEX Zhang et al. (2023b) sequentially expands a policy set to accommodate the change in transfer learning, while our work focuses on a parallel policy set in online RL. Our work is also related to population-based methods (Jung et al., 2020; Parker-Holder et al., 2020). DiCE (Peng et al., 2020) is a work related to ours which adds a diversity regularization to improve the diversity in an ensemble. MED-RL (Sheikh et al., 2022) further introduced a set of regularization methods to prevent the collapse of representations in ensemble RL. Our setting is also similar to the Thompson Sampling RL, which maintains a distribution of policies and samples a set of policies from the posterior for exploration, and sync after interactions (Osband & Roy, 2017; Chen et al., 2022).

**Exploration with Intrinsic Rewards**. In this work, we leverage the proposed *Collaborative Reward Generator* to achieve collaborative exploration via a collaborative intrinsic reward (Pathak et al., 2017; Burda et al., 2019; Badia et al., 2020; Zhang et al., 2020b; Seo et al., 2021; Jo et al., 2022;

Wang et al., 2023b; Yuan et al., 2023; Zahavy et al., 2023). The key point is to share the novelty information among agents to improve the exploration efficiency from a holistic view. In principle, the idea of collaborative exploration is agnostic to the selected intrinsic reward method. For example, the CE variant uses a KNN-based intrinsic reward (Seo et al., 2021) and the CE-RND variant uses a RND-based intrinsic reward (Burda et al., 2019). In this work, we mainly focus on demonstrating the effectiveness of the idea of collaborative exploration in a parallel-agent setting, and we leave the comparison of more intrinsic reward methods for future work.

## B  LIMITATIONS AND FUTURE DIRECTIONS

A major limitation of our work is the increasing computational cost. Given an agent set $\mathbb{A}$ of size $N$, our model is roughly $N$ times larger than a single RL agent. Moreover, the proposed method requires learning an extra contrastive encoder or RND modules to compute the intrinsic rewards. Therefore, our method takes a longer training time than its single RL agent counterpart. In this work, we attempt to mitigate this issue by implementing our code in JAX with vectorized model architectures. We provide more detailed discussions about the problem of time efficiency in the Appendix D.6.

In this work, we investigate how to achieve collaborative exploration in an agent set $\mathbb{A}$ via sharing different information. Moreover, we treat all the other agents in $\mathbb{A}$ equally when we compute the collaborative intrinsic rewards for the $i$-th agent. An interesting future direction is to assign different weights to each agent when we compute the collaborative intrinsic rewards. For example, we can use each agent's recent performance or uncertainty as the weight to compute a more fine-grained reward. Another interesting future direction is to use a policy distillation stage to extract a distilled policy from the agent set. In addition, the samples in the shared buffer could be quite off-policy for each agent, where many of them are collected by the other agents. Therefore, it would be interesting to investigate the combination of more advanced off-policy policy evaluation methods.

## C  EXPERIMENT DETAILS

In this section, we provide more details on the experiment setups.

### C.1  EXPERIMENT SETUPS

In the experiment, we compare the baseline agents on the DeepMind Control suite (DMC) benchmark tasks (Tassa et al., 2018). In all experiments, we run the RL agents for 1 million environment steps for each baseline except for the A2C and DIAYN. Since A2C is more close to an on-policy algorithm, where the collected samples are discarded once used. We run A2C for 1e7 environmental steps. We run DIAYN for 2e6 environmental steps, where the first 1e6 steps are used for pre-training skills and the second 1e6 steps are used for tine-tuning. For the proposed CE method, we use 4 agents in the state-based experiments and use 2 agents in the pixel-based experiments. Agents interact with the environment for the same environment steps, *i.e.*, each agent in the state-based experiments interacts with its own environment for 2.5e5 environment steps. In addition, each agent takes 4 gradient updates per environment step to match the 1e6 gradient steps for each agent as in the other baselines.

Table C.1 shows the hyperparameters we use in the experiments. For the proposed components for the collaborative exploration, we use an epsilon $\epsilon$ of 0.2 for the collaborative exploration and the sampled action number $M = 10$. We use $K = 20$ for the KNN cosine similarity in the contrastive collaborative encoder. To approximate the agent's behaviors, we use a memory bank of size 300 to store the agent's recent trajectories. Specifically, the memory bank is a FIFO buffer that stores the agent's latest trajectories. Since two consecutive states $s_t$ and $s_{t+1}$ are usually very similar in different tasks, we further introduce a skip frame parameter of 5 to only add one state into the memory bank for every 5 environment steps. To construct the positive samples to train the contrastive encoder, we adopt nearby samples $\{s_{t-L}, \cdots . s_{t-1}, s_{t+1}, \cdots , s_{t+L}\}$ as the positive samples for the state $s_t$ with the $L = 5$. All the other samples are viewed as the negative samples for $s_t$, and in the experiment, we use the other positive samples in the batch as the negative samples. Figure 10 illustrates the overall training pipeline of the proposed CE method. We implement the baselines in JAX (Bradbury et al., 2018), and we adopt the parameter settings from the JaxRL2 library (Kostrikov,

Table 4: **Collaborative exploration parameters**: some parameters for the proposed $\epsilon$-collaborative exploration, collaborative training, and collaborative evaluation.

| Hyperparameter | DMC15 |
|---|---|
| $\epsilon$ for the $\epsilon$-collaborative exploration | 0.2 |
| Sampled action number $M$ | 10 |
| Memory skip | 5 |
| Memory bank size | 300 |
| Positive sample horizon | 10 |
| KNN | 20 |
| Batch size | 256 |
| Intrinsic reward weight $\lambda$ | 0.2 |
| Temperature $\tau$ in contrastive loss | 0.07 |

2022). More model parameters are summarized in the Table 5. Algorithm 1 shows the pseudocode for the proposed method.

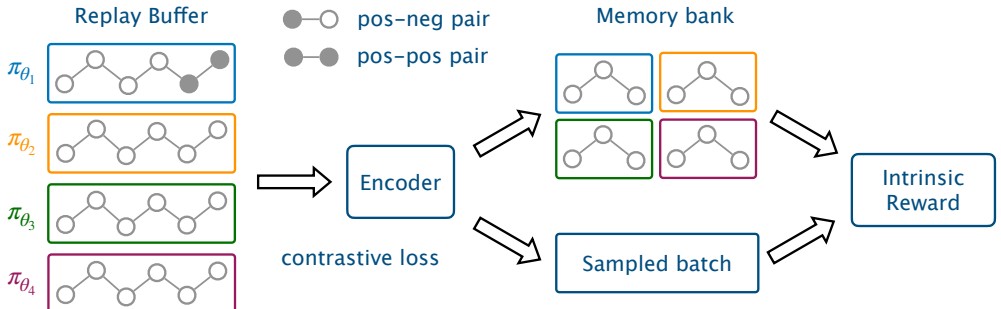

Figure 10: **An illustration of the training pipeline**: similar to ATC (Stooke et al., 2021), we select nearby transitions as positive samples and train the contrastive encoder with InfoNCE loss.

---

**Algorithm 1** Contrastive Collaborative Exploration

**Input:** total environment step $T$, evaluation frequency $F$, agent number $N$, shared buffer $\mathcal{D}$, memory bank $\mathcal{M}$, action number $M$ for the $\epsilon$-collaborative exploration.
**Initialize:** the policy the $\pi_i$ and value function $Q^{\pi_i}(s, a)$ for each SAC agent, timestep $t = 0$.
**while** $t <= T$ **do**
    **for** $i = 1$ **to** $N$ **do**
        The $i$-th agent sample an action $a_t$ according to the $\epsilon$-collaborative exploration (Eqn. 5).
        Store the sampled transition $(s_t, a_t, r_t, s_{t+1})$ to the shared buffer $\mathcal{D}$ and memory bank $\mathcal{M}$.
        $t = t + 1$.
    **end for**
    Sample data from the memory bank $\mathcal{M}$ to compute the intrinsic reward (Eqn. 3).
    Update the backbone SAC agent.
    Update the contrastive encoder with Eqn. 4.
    **if** $t\%F == 0$ **then**
        Collaborative evaluation using the softmax policy.
    **end if**
**end while**

---

# D    ADDITIONAL EXPERIMENT RESULTS

## D.1    EVALUATION CURVES ON THE DMC15 BENCHMARK TASKS

Figure 11 shows the evaluation curves for each baseline on 15 DeepMind Control suite (DMC) tasks . We report the mean and standard deviation of the evaluation scores across 5 random seeds. We use

Table 5: **Model parameters**: some other parameters we use in the experiments.

| Hyperparameter | DMC15 |
|---|---|
| Actor network | 256-256 |
| Critic network | 256-256 |
| RND network | 256-256 |
| Encoder network | 64-64 |
| Batch size | 256 |
| Learning rate | 3e-4 |
| Replay buffer size | 1e6 |
| Discount factor | 0.99 |
| Ensemble size | 4 |
| Actor gradient clip | 20 |
| Critic gradient clip | 20 |
| Target entropy | $-0.5 * |\mathcal{A}|$ |

the mean of the last 10 evaluation results, where the agent is evaluated for every 5000 timesteps, after 9.5e5 timesteps as the evaluation score for one random seed. To facilitate visualization, we further smooth the curves with a sliding window of length 10. We can observe that the proposed CE method outperformed or matched other baselines in most of the tasks.

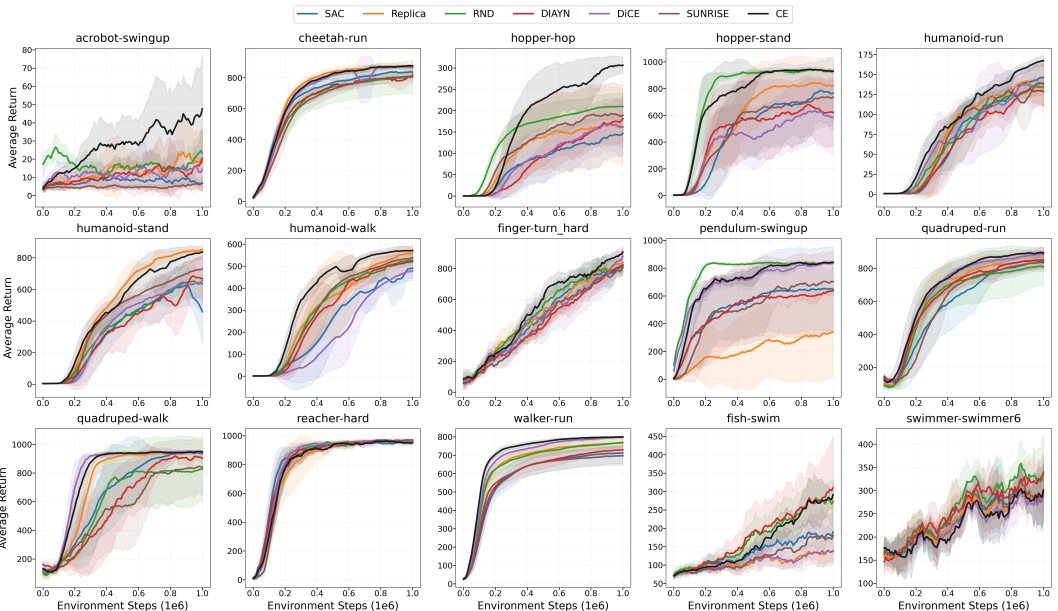

Figure 11: **Evaluation Curves on the DMC15 Tasks**: we report the average and standard deviation of the evaluation scores across 5 random seeds.

## D.2 ABLATION STUDIES ON DIFFERENT PROPOSED COMPONENTS

In this subsection, we investigate the combination of the different proposed components – the collaborative data collection (Col-Unroll), collaborative training (Col-Training) and collaborative evaluation (Col-Eval). For the collaborative evaluation, we simply use the softmax-based mixture policy. We also add the results of the Replica baseline and CE for the comparison. The combination of all three components is denoted as the *All* method. Table 6 shows the results of four DMC tasks. We can observe that combining the three components performs the best except for the *fish-swim* task, where using the collaborative evaluation is effective and using the collaborative data collection hurts the performance. This is because the *fish-swim* is a challenging goal-reaching task, and adding the collaborative data collection might distract the agent from reaching the target goal when some other agents sample sub-optimal actions. Overall, removing the collaborative training (w/o Col-Training)

performs the worst, which indicates that collaborative training via the proposed collaborative reward generator is the most effective component.

Table 6: **Ablation of different components:** collaborative training is the most important component.

| Env | Replica | CE | All | w/o Col-Eval | w/o Col-Unroll | w/o Col-Training |
|---|---|---|---|---|---|---|
| fish-swim | 152.4 (30.6) | 305.2 (15.4) | 330.2 (25.8) | 222.8 (54.6) | **377.5 (71.0)** | 207.3 (35.1) |
| hopper-hop | 164.6 (93.3) | 306.8 (20.6) | **343.1 (61.5)** | 333.7 (62.2) | 325.4 (81.4) | 162.6 (60.1) |
| humanoid-walk | 561.7 (32.7) | 574.6 (20.2) | **589.3 (17.1)** | 573.3 (13.4) | 558.3 (17.3) | 517.8 (68.7) |
| pendulum-swingup | 345.0 (320.8) | 841.4 (4.3) | **843.6 (5.4)** | 841.8 (3.8) | 841.4 (3.2) | 661.1 (312.8) |

## D.3 ABLATION STUDIES ON AGENT NUMBERS

In this subsection, we investigate the effect of using different agent numbers in the agent set $\mathbb{A}$. Table 7 compares the results of using 2/4/8/16/32/64 agents in the Replica baseline and the CE method. We can observe that the performance of scaling to more agents actually depends on the tasks. It's notable that there is a trade-off between policy diversity and training stability. With more agents, we have more diverse policies and help to solve tasks where exploration is particularly difficult, i.e., the goal-reaching fish-swim task. On the other hand, the off-policyness of the sampled data increases as we have more agents. Given a sampled batch, there are only on average $1/N$ samples that are collected by each agent. As the number $N$ increase, the problem becomes closer to an offline RL setting, where the optimization becomes more challenging, i.e., the performance variances increase significantly for larger $N$. The focus of this work is to illustrate the effectiveness of collaborative exploration among agents, and we leave the optimization challenge for future work. Therefore, using a medium size of agent set strikes a balance of exploration and optimization.

Table 7: **Ablation of agent numbers:** a larger number of agents could improve the diversities and help exploration, but it also makes the optimization more difficult due to more off-policy data.

| | fish-swim | hopper-hop | humanoid-walk | pendulum-swingup |
|---|---|---|---|---|
| Replica 2 | 156.9 (45.6) | 166.8 (78.7) | 509.5 (42.1) | 365.1 (263.6) |
| Replica 4 | 152.4 (30.6) | 164.6 (93.3) | 561.7 (32.7) | 345.0 (320.8) |
| Replica 8 | 260.6 (68.7) | 224.8 (51.3) | 438.1 (77.5) | 369.0 (351.4) |
| Replica 16 | 414.3 (58.6) | 223.8 (40.3) | 567.9 (73.1) | 330.1 (328.0) |
| Replica 32 | 565.8 (65.6) | 160.2 (56.7) | 578.5 (88.3) | 276.6 (327.0) |
| Replica 64 | 569.5 (89.1) | 152.7 (60.1) | 432.3 (138.1) | 288.9 (279.2) |
| CE 2 | 324.7 (28.5) | 207.2 (129.7) | 483.0 (79.2) | 837.2 (9.2) |
| CE 4 | 305.2 (15.4) | **306.8 (20.6)** | **574.6 (20.2)** | **841.4 (4.3)** |
| CE 8 | 360.1 (68.2) | 205.0 (37.6) | 523.1 (41.6) | 830.2 (14.1) |
| CE 16 | 458.6 (76.9) | 232.4 (60.5) | 531.0 (51.7) | 825.1 (18.1) |
| CE 32 | **604.6 (83.5)** | 226.8 (70.7) | 545.2 (67.8) | 824.5 (23.5) |
| CE 64 | 601.6 (103.7) | 201.2 (86.2) | 518.2 (129.0) | 828.4 (26.3) |

## D.4 ABLATION STUDIES ON THE CONTRASTIVE ENCODER

One key point in our method is to measure the similarity between the sampled transitions to each agent's behaviors. To this end, we need one module to extract the representations for each transition and compute the similarity. In practice, we could select any suitable representation learning methods to measure the similarities between transitions. In this subsection, we compare different encoders for the CE method. In particular, we compare the proposed contrastive encoder to an identical encoder, which outputs the raw observation, and a random encoder (Seo et al., 2021). From the results in Table 8, we can observe that using either the identical encoder or the random encoder performs much worse than the proposed contrastive encoder in the CE. These results first prove the effectiveness of the proposed contrastive encoder and highlight the importance of learning a good representation for the similarity computation.

Table 8: **Ablation of different encoders:** contrastive encoder can learn useful representations.

| Env | Replica | CE | Identical | Random |
|---|---|---|---|---|
| fish-swim | 152.4 (30.6) | **305.2 (15.4)** | 176.4 (49.4) | 190.2 (61.3) |
| hopper-hop | 164.6 (93.3) | **306.8 (20.6)** | 147.7 (63.4) | 179.7 (63.6) |
| humanoid-walk | 561.7 (32.7) | **574.6 (20.2)** | 453.8 (131.4) | 537.3 (26.6) |
| pendulum-swingup | 345.0 (320.8) | **841.4 (4.3)** | 697.5 (259.5) | 514.6 (361.5) |

## D.5 ABLATION STUDIES ON THE KNN DISTANCE

In this subsection, we study the effects of using different $K$ values for the KNN distance. Firstly, we want to point out that the $K$ parameter is closely related to the intrinsic reward weight $\lambda$:

$$\hat{r}^i(s, a) = r_{\text{task}}(s, a) + \lambda r^i_{\text{intrinsic}}(s, a).$$

As we use the negative KNN cosine similarity as the intrinsic reward $r^i_{\text{intrinsic}}(s, a)$, so a larger $K$ value corresponds to a smaller cosine similarity (larger intrinsic reward). The overall effect of the intrinsic reward depends on both the $\lambda$ and $K$ parameters. Therefore, to investigate the influence on the parameter $K$, we use a fixed $\lambda = 0.2$ in the following experiments. Table 9 shows the results of using different $K$ as the KNN cosine similarity, and we can observe that using a medium value of $K = 20$ performs the best. When we select a small value, *i.e.*, $K = 5$, then the cosine similarity is likely to be always close to 1 and the intrinsic reward degrades to a constant reward shifting. On the other hand, when we select a large value, *i.e.*, $K = 30$, then the intrinsic reward would have a large value which might distract the agent from the real task reward.

Table 9: **Ablation of KNN:** a medium value of $K = 20$ performs the best.

| Env | Replica | K=5 | K=10 | K=20 | K=30 |
|---|---|---|---|---|---|
| fish-swim | 152.4 (30.6) | 172.8 (79.2) | 210.6 (77.4) | **305.2 (15.4)** | 241.8 (49.8) |
| hopper-hop | 164.6 (93.3) | 165.4 (64.0) | 264.7 (95.1) | **306.8 (20.6)** | 256.7 (50.3) |
| humanoid-walk | 561.7 (32.7) | 539.7 (29.6) | 545.0 (26.6) | 574.6 (20.2) | **577.0 (19.9)** |
| pendulum-swingup | 345.0 (320.8) | 773.8 (135.3) | 839.4 (2.8) | **841.4 (4.3)** | 837.1 (5.3) |

## D.6 COMPARISON OF TIME EFFICIENCY

Compared to the single RL agent baseline, the proposed method needs to learn more RL agents, hence leading to a longer training time. To mitigate this issue, we implement our code in JAX (Bradbury et al., 2018) with vectorized model structures (Flajolet et al., 2022). Overall, the CCE agent with an ensemble size of four takes around 2 times slower than a single JAX-based SAC agent. More detailed running time for each task can be referred to Table 10. Here, we compare to a single SAC agent which takes more gradient updates per step (UPS), which roughly matches the wall-clock running time of our method. As we can observe in the Figure 12, using more gradient updates can usually improve the performance (Nikishin et al., 2022). The performance gap between SAC (UPS3) with our method indicates the benefits of incorporating the collaborative exploration information.

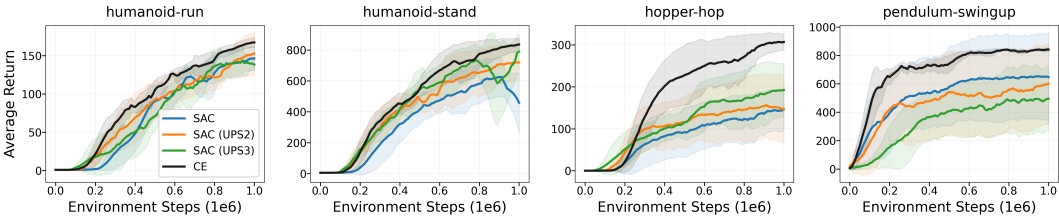

Figure 12: **SAC with more gradient updates:** running SAC with more gradients per update.

Table 10: **A comparison of wall-clock running time on different DMC tasks:** our method using an ensemble of 4 agents is roughly 2 times slower than a single JAX-based SAC agent. The running time is measured on a machine with RTX 3090 GPU and Intel i9-12900KF CPU.

| Environment | State Dimension $|\mathcal{S}|$ | Action Dimension $|\mathcal{A}|$ | SAC (min) | Ours (min) |
|---|---|---|---|---|
| acrobot-swingup | 6 | 1 | 23 | 45 |
| cheetah-run | 17 | 6 | 23 | 45 |
| finger-turn-hard | 12 | 2 | 25 | 50 |
| fish-swim | 24 | 5 | 26 | 51 |
| hopper-hop | 15 | 4 | 24 | 47 |
| hopper-stand | 15 | 4 | 27 | 55 |
| humanoid-run | 67 | 21 | 36 | 75 |
| humanoid-stand | 67 | 21 | 36 | 75 |
| humanoid-walk | 67 | 21 | 37 | 75 |
| pendulum-swingup | 3 | 1 | 24 | 46 |
| quadruped-run | 78 | 12 | 32 | 62 |
| quadruped-walk | 78 | 12 | 33 | 62 |
| reacher-hard | 6 | 2 | 23 | 45 |
| swimmer-swimmer6 | 25 | 5 | 37 | 72 |
| walker-run | 24 | 6 | 28 | 55 |

## D.7 EXPERIMENTS ON THE ATARI GAMES

We further compare the proposed method to DQN on the Asterix, BeamRider, Breakout, and SpaceInvaders environments. In each environment, we train the agent for 5e6 environment steps, and we evaluate the agent for every 2e5 steps. We run for 5 random seeds and report the mean evaluation score and standard deviation. We further smoothed the curves with a sliding window of 5. Results are shown in the Figure D.6. We can observe that CE generally outperforms the DQN baseline.

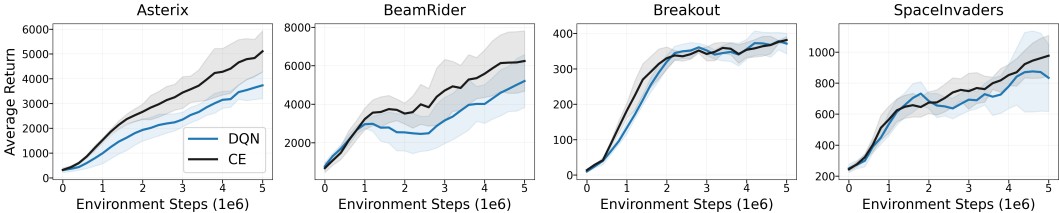

Figure 13: **Results on the Atari games:** CE outperforms DQN on the selected tasks.

## D.8 COLLABORATIVE RND DETAILS

Here we further provide an illustration of the proposed collaborative variant using RND (RND+CE) in the Figure 14. In RND+Replica, each agent is trained with its own data. In RND+CE, it learns a shared RND module using the shared buffer, containing data from all the agents.

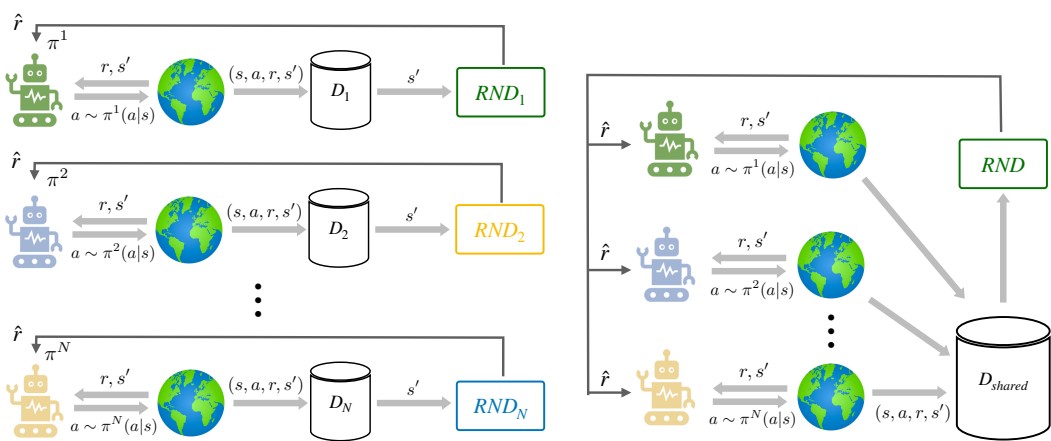

Figure 14: **An example of collaborative novelty using RND:** (Left) RND+Replica agents training with its own data; (Right) RND+CE: collaborative RND agents training with a shared RND module.

