# OpenReview forum: "A Collaborative Perspective on Exploration in Reinforcement Learning"
_ICLR.cc/2024/Conference — Submitted to ICLR 2024_

### Official Review · Reviewer_6JRU · 2023-10-21

**Soundness:** 3 good
**Presentation:** 3 good
**Contribution:** 3 good
**Rating:** 6
**Confidence:** 4

**Summary:**

Built on a novel collaborative perspective on exploration, this paper proposes a simple yet powerful extension for improving exploration in reinforcement learning. Overall, this paper is well presented and shows promising results via thorough experiments. I only have few questions related to details of the algorithm design and implementation.

**Strengths:**

(a) This paper proposes to study the exploration problem in (single-agent) RL from a collaborative perspective, which is quite novel and well motivated.

(b) The algorithm design is simple and well-presented, of which the effectiveness is well supported by solid empirical results.

**Weaknesses:**

(a) Maintaining $N$ training agents and environments at the same time may increase the computation and space complexity.

(b) The related work part can be further extended.

**Questions:**

(a) Is this the first work on collaborative exploration in (single-agent) deep RL?

(b) Do the $N$ agents share parameters in their policy or value networks (I guess not, then all agents need to be trained)?

(c) Could you explain why the $L_2$ norm of the action difference is a good measure for behavior diversity, rather than some other designs (like difference in $\pi_i(\cdot|s_t)$)?

(d) It would be great if the authors can release the codes for the baseline algorithms as well. Details on reproducing each figure (or training process) in the paper should also be given in the readme file.

---

> ### Author Response · Authors · 2023-11-22
> **Reply to Reviewer 6JRU**
>
> We thank the reviewer for the constructive review and insightful comments. Responses to the questions are below:
>
> > Q1: maintaining N agents and environments at the same time may increase the computation and space complexity.
>
> We admit that the increasing computation and space complexity is the major limitation of our work as we explained in the common question section. In order to mitigate this issue, we implement our code in JAX with vectorized model architectures. In the experiment, using an agent number of size 4, the wall clock running time is around 2 times of training a single agent.
> Another potential solution is to use a multi-head version of this algorithm, which is exactly the following question as we will discuss later.
>
> > Q2: The related work part can be further extended.
>
> Thanks for the suggestion. We rewrote the related work section.
>
> > Q3: Is this the first work on collaborative exploration in (single-agent) deep RL?
>
> No, we are not the first work on this topic. For example, one of the selected baselines, DiCE, also studies this problem by adding a diversity regularization to the loss function. However, our discussions of how and when to collaborate among agents and the instantiation of using the collaborative reward are novel.
>
> > Q4: Do the N agents share parameters in their policy or value networks?
>
> No, the current agents do not share parameters. Since the main focus of this work is to study how and when to collaborate among agents, we adopt the most basic model architectures in the experiments.
> Using a multi-head variant of the proposed method would be easy to implement where each agent shares the same torso network and uses different heads as the policy/value function.
> Experimental results of the multi-head variant are reported in Section 5.8.
>
>
> > Q5: Why is the $L_2$ norm of action difference a good measure for behavioural diversity?
>
> We adopt the L2 norm in the pervious experiments mainly because it is intuitive and the overall results are similar to other more complex metrics. Since the purpose of Table 3 is mainly for providing some qualitative understanding of the proposed method. Therefore, we select the MSE-based metric due to its simplicity.
> We followed the reviewers' suggestions and tried some other metrics, i.e., mutual information $I(S;A)$ and KL divergence $D_{KL}(\pi_i, \pi_j)$ to approximate policy diversity in the updated paper.
>
> > Q6: It would be great if the authors can release the codes for the baseline algorithms as well
>
> We have submitted the code of the proposed method in supplementary material at the time of our original submission.
> Following your suggestion, we will release more code including some baselines, the visualization scripts etc by including a Github link in the final version.

---

### Official Review · Reviewer_AU7q · 2023-11-01

**Soundness:** 2 fair
**Presentation:** 2 fair
**Contribution:** 2 fair
**Rating:** 5
**Confidence:** 4

**Summary:**

The paper proposes a collaborative approach to improving exploration in reinforcement learning (RL). Rather than using a single agent, it introduces multiple agents that interact with separate environments in parallel. The key ideas are:

- Multi-agent formulation: Maintain a set of N parallel agents, each with its own policy and environment. All agents share a common replay buffer.
- Collaborative reward generator: Calculate an intrinsic reward for each agent that encourages it to visit novel states not explored by other agents. This induces collaboration and specialization between agents.
- Collaborative data collection: Agents can also explicitly coordinate during data collection, e.g. selecting diverse actions compared to others.

The approach is evaluated on DM Control tasks. Results show improved exploration and performance compared to single agent baselines and other intrinsic reward methods like RND and ICM. The idea of collaborative reward is general and can be integrated with different algorithms.

**Strengths:**

- Intuitive idea of converting single agent RL to collaborative setting to improve exploration.
- Flexible framework that can work with different intrinsic reward designs.
- Showcases benefits over reasonable baselines like RND and ICM.
- Evaluated on established DMC benchmark tasks.
- Collaborative rewards induce specialization between agents and
- General idea that can be integrated with many RL algorithms.

**Weaknesses:**

Some of the obvious issues like increase in computation cost proportional to number of agents as well as training wall-clock time should be noted in the main paper itself. Overall given about 2x increase in computational cost, the minor improvements in performance don't seem _that_ significant and likely achievable with standard methods trained for as long (with appropriate hyperparameters).

Connections with parameter sharing in MARL literature (for example [1, 2, 3, 4]) are completely missing. The general setup of using environment experience from 'multiple' agents to train a single policy is also used in those contexts and connections with explorations as investigated here might be relevant in those contexts as well. In general the literature review seems too biased towards just the last couple years of RL papers.

[1] https://arxiv.org/abs/2005.13625

[2] https://arxiv.org/abs/2102.07475

[3] https://link.springer.com/chapter/10.1007/978-3-319-71682-4_5

[4] https://openreview.net/forum?id=YVXaxB6L2Pl

**Questions:**

- How does contrastive reward work in early parts of training when initial policies are mostly random?
- How well would it scale to different kind of environemnts like Atari?
- Any intuitions on how to balance intrinsic vs task rewards?

---

> ### Author Response · Authors · 2023-11-22
> **Reply to Reviewer AU7q**
>
> We thank the reviewer for the constructive review and insightful comments. Responses to the questions are below:
>
> > Q1: increase in computation cost and minor performance improvements
>
> Thanks for the suggestion. We added a description of the running time in Section 5.2.
> Due to the inherent non-determinsticness of RL, we admit that the proposed method sometimes only achieves minor performance improvements. This usually occurs on simple tasks., i.e., cheetah-run and reacher-hard in DMC, where a single agent can easily solve the task. The benefits of using collaborative information are more significant in some tasks where exploration is more difficult, i.e., hopper-hop and humanoid-run.  In these tasks, a single agent might get stuck in sub-optimal behaviors while leveraging the collaborative information is a simple and robust method to improve the performance.
> Due to the variation caused by tasks, it is better to compare the overall performance across a set of tasks. In Figure 3, we report the aggregated performance over 15 DMC tasks.
>
> Furthermore, Figure 11 in the Appendix D.6 shows the results of running a single agent for more wall-clock running time. We can observe that a single agent still performs worse when it trains for a similar wall-clock running time.
>
>
> > Q2: connections with parameter sharing in MARL literature
>
> Thanks for the reminder. We added discussions of the connections and differences of our work with parameter sharing in MARL in both Section 2 and Appendix A. We also added a multihead variant of CE in Section 5.8, which uses parameter-sharing.
>
> > Q3: How does contrastive reward work in early parts of training when initial policies are mostly random?
>
> In the early parts of training:
> 1. When initial policies are mostly random, different agents usually take low-quality actions and receive near zero rewards from the environment.
> 2. The introduced collaborative intrinsic reward encourages the agents to take actions that are not only novel to itself but also to the other agents. (Please refer to the reply of Q1 to R2). Actually, this is related to your other question on the balancing between intrinsic and task rewards.
> 3. When some of the agents sampled good actions and received high rewards, then this information is shared to the other agents via the shared replay buffer.
>
>
> > Q4: How well would it scale to different kinds of environments like Atari?
>
> We added extra experiments on the Atari games in Appendix D.7. We compare the proposed method to DQN on the Asterix, BeamRider, Breakout, and SpaceInvaders environments. In each environment, we train the agent for 5e6 environmental steps, and we evaluate the agent for every 2e5 steps. We run for 5 random seeds and report the mean evaluation score and standard deviation. We can observe that the proposed CE method also achieves good performances in Atari tasks.
>
> > Q5: Any intuitions on how to balance intrinsic vs task rewards?
>
> - In the current algorithm, as the quality of sampled data increases, the scale of the task will gradually outweigh the intrinsic reward.
> - A widely used practice is to linearly decay the intrinsic reward coefficient during the training.
> - Or we can adopt some more advanced method [1] to tune the intrinsic reward weight automatically.
>
> [1] Yuan, Mingqi, et al. "Automatic Intrinsic Reward Shaping for Exploration in Deep Reinforcement Learning." arXiv preprint arXiv:2301.10886 (2023).

---

### Official Review · Reviewer_JJ8w · 2023-11-03

**Soundness:** 3 good
**Presentation:** 3 good
**Contribution:** 3 good
**Rating:** 5
**Confidence:** 4

**Summary:**

This paper presents collaborative exploration a framework for multi agent exploration in reinforcement learning. In this framework each agent interacts with its copy of the environment and all agents share information. The authors propose an exploration bonus that is aware of other agents to incentivize every agent to visit unexplored locations. The bonus can be implemented using a constrative reward or using RND. The proposed method is then evaluated on DMC15 tasks including pixel based tasks and is shown to perform better the baselines.

**Strengths:**

The paper takes on the interesting topic of exploration in reinforcement learning. While single-agent scenarios have received more attention, the multi-agent setting remains a bit of an unexplored territory, even though it holds promise for substantial improvements.
The paper is well-written and easy to follow. The method they suggest is put through its paces across various tasks, and it consistently shows better results.
The visual aids in Figures 5 and 6 are appreciated and help us get a better grasp of how the proposed algorithm works.

**Weaknesses:**

The paper uses the term collaborative exploration when this subject has been quite studied in the past under the name "concurrent exploration". The paper is also missing a lot of existing work in this area [1-6]. I think it is worth adding these references in the next revision of the paper as section 3.1 could seems like the author came with concurrent exploration.

It seems like the time complexity for computing the embeddings in section 4.1 is linear in the number of agents and number of states in the replay buffer. This seems detrimental, I assume that we're interested in using a multi agent method because we care more about wall clock time than sample complexity, in that case the additional cost to compute the embedding may not be worth it. Table 10 shows that the wall clock times doubles with collaborative exploration.


[1] Concurrent Reinforcement Learning from Customer Interactions, Silver et al. ICML 2013
[2] Coordinated Exploration in Concurrent Reinforcement Learning, Dimakopoulou et al., ICML 2018
[3] Scalable Coordinated Exploration in Concurrent Reinforcement Learning, Dimakopoulou et al., NeurIPS 2018
[4] Regret Bounds of Concurrent Thompson Sampling, Chen et al. NeurIPS 2022
[5] Efficient PAC-Optimal Exploration in Concurrent, Continuous State MDPs with Delayed Updates, Pazis
[6] Introducing coordination in concurrent reinforcement learning, ICLR 2022 workshop

**Questions:**

"Eqn.(5) encourages the agent to visit states that are not only novel from its own perspective, but also
to respect other agents’ intrinsic motivation in pursuit of novelty."
If I understand Eqn (5) correctly it is pushing agent i towards states it has not been but also towards states that have been visited by other agents?

"Simply, we can use the softmax function as the classifier" (section 4.2) and " In the following experiments, we always select the first agent for evaluation." (section 5).
Does it mean that only collaborative exploration is able to use the softmax action selection. This might be important as CE implicitly uses a higher capacity policy.

An ablation study would have helpful to understand the impact of each component of collaborative exploration, particularly the sofmax action selection procedure.

In section 5.3 / Figure 4, could you add RND?

On Figure 5 why do we still more trajectories going towards the bottom? Shouldn't it be optimal to split and the two goals equally?

"In the experiment, we use λ = 1 for the UCB method, ϵ = 0.2 and M = 10 for the ϵ-collaborative method. ϵ-greedy method also uses ϵ = 0.2" can you explain how hyperparameters were tuned?

The number of agents used for experiments (2 - 8) is quite low, could you add some results with 32, 64, 128? It would be interesting to see how the methods scale with the number of agents. It is possible that the reward becomes too noisy and lead to worse results. It also likely explains why CE4 does better than CE8 in Table 7.

---

> ### Author Response · Authors · 2023-11-22
> **Reply to Reviewer JJ8w [Part I]**
>
> We thank the reviewer for the constructive review and insightful comments. Responses to the questions are below:
>
> > Q1: missing related work on concurrent exploration.
>
> Thanks for the suggestion. We first added a paragraph dedicated to concurrent exploration in the related work section and also explain the connection between our work and concurrent exploration in Section 3.1.
>
>
> > Q2: time complexity for computing the embeddings is linear in the number of agents and number of states in the replay buffer.
>
> Yes.  Denote the memory buffer size as M, embedding dimension as d, and the agent number as N. Then, the time complexity to compute the KNN intrinsic reward is $O(KNMd)$.
> To mitigate this issue, we use a memory buffer for each agent with a relatively small size (M = 300).
> - In practice, as discussed in the response to Q7 for R1, $N$ does not need to be a large number to strike the balance between complexity and benefits brought to exploration.
>
> > Q3: meaning of intrinsic reward
>
> From the definition of the intrinsic reward, we can observe that it is composed by two parts:
>
> - ego part: encourages the agent to take action to visit states that are less visited by itself.
> - mutual part: encourages the agent to take action to visit states that are less visited by the other agents.
>
> We use a reward weight to trade-off these two parts. Overall, this intrinsic reward encourages agents to take different behaviours and visit less visited states from a holistic perspective.
>
> > Q4: only collaborative exploration is able to use the softmax action selection
>
> The proposed collaborative data collection and collaborative evaluation are optional and only used in Section 5.6 and Section 5.7.
> In Section 5.2/5.3/5.4, we always select the first agent for evaluation to validate the effectiveness of the proposed collaborative intrinsic reward.  We use the softmax action selection in Section 5.7 to validate the effectiveness of using collaborative information in the evaluation phase. Moreover, the softmax action selection is just a simple example to raise the point that using collaborative information in the evaluation is effective. More complex algorithmic designs can be used as well, which we leave for a future work.

---

> ### Author Response · Authors · 2023-11-22
> **Reply to Reviewer JJ8w [Part II]**
>
> > Q5: Ablation study for each component
>
> Thanks for the suggestion. We provide the ablation study for each component in Appendix D.2. In Table 6, we can observe that combining the three components performs the best except for the fish-swim task, where using the collaborative evaluation is effective and using the collaborative data collection hurts the performance. This is because the fish-swim is a challenging goal-reaching task, and adding the collaborative data collection might distract the agent from reaching the target goal when some other agents sample sub-optimal actions. Overall, removing the collaborative training (w/o Col-Training) performs the worst, which indicates that collaborative training via the proposed collaborative reward generator is the most effective component.
>
> > Q6: In section 5.3 / Figure 4, could you add RND?
>
> Thanks for the suggestion. We have added the RND in Figure 4. The proposed CE method also outperforms RND in the visual task experiments.
>
> > Q7: Why are more trajectories going to the bottom in Figure 5?
>
> It's mainly due to the randomness inherent in the agent's data unrolling and how we sample the trajectories for visualization. When the agent is well trained, we will on average have nearly the same number of trajectories towards the upper goal and the bottom goal.
> In the previous experiments, we have chosen to plot one trajectory for every 20 trajectories to avoid cluttered plots. However, this 1/20 ratio might be a bit too large, which can lead to sampling bias and make the figure mis-leading.
> If we plot more frequently, i.e., one trajectory for every 10 trajectories, then trajectory distribution will be more balanced. We have updated Figure 5 in the revised paper accordingly.
>
> >  Q8: How parameters are selected
>
> Mainly following some previous settings from the SUNRISE paper and did not tune much.
> Better results are possible with more tuning.
>
> > Q9: More agent numbers 32/64/128
>
> Thanks for the suggestion. We added results of using 16/32/64 agents in Table 7 in Appendix D.3. The performance of scaling to more agents actually depends on the tasks.
> It's notable that there is a trade-off between policy diversity and training stability:
> - With more agents, we have more diverse policies and help to solve tasks where exploration is particularly difficult, i.e., the goal-reaching fish-swim task.
> - On the other hand, the off-policyness of the sampled data increases as we have more agents. Given a sampled batch, there are only on average 1/N samples that are collected by each agent. As the number N increase, the problem becomes closer to an offline RL setting, where the optimization becomes more challenging, i.e., the performance variances increase significantly for larger N.
> For the proposed CE agents, it is also indeed possible that the reward becomes too noisy and lead to worse results. Therefore, using a medium size agent set strikes a good balance of exploration and optimization.

---

### Official Review · Reviewer_ArFd · 2023-11-06

**Soundness:** 1 poor
**Presentation:** 2 fair
**Contribution:** 1 poor
**Rating:** 3
**Confidence:** 5

**Summary:**

Authors present a parallelized training algorithm for a single agent in view of pooling experience samples for exploration. The proposed method is tested on DeepMind Control Suite against several baselines across multiple scenarios.

**Strengths:**

Originality
It is hard to assess the originality of the submission; the proposed method is not clearly distinguished from A3C, Ape-X, or IMPALA, save a brief mention in Appendix A.

Quality
A variety of evaluation setups and baselines have been incorporated. Presented figures and tables clearly indicate a superior performance in most setups.

Clarity
Figure 2 serves as an apt visual abstract for the proposed method. Reproducibility efforts have been included where applicable.

Significance
It is difficult to gauge whether or how the algorithm works well, given that its special design oriented towards better exploration still underperforms Replica in HalfCheetah-V3 in terms of the variance proxy and that no further explanation is provided as to how that might be the case.

**Weaknesses:**

First of all, the “multi-agent” phrasing is misleading. Nowhere in A3C, Ape-X, or IMPALA is the term used to refer to parallel instantiations of one agent. Repurposing a well-established terminology should be accompanied by far more solid evidence than a mere footnote. Related works are poorly taxonomized. For instance, if MARL research and the proposed method are indeed “very closely related”, how is it that no MARL algorithm is tested against?
Important works on diversity, such as Diversity Is All You Need, are not discussed, and no comparison is made against information-theoretic classes of diversity-objective RL algorithms.
Despite admitting a resemblance with distributed RL, none of the cited algorithms is set up for comparative evaluation.
There is no justification as to how the variance proxy may be a better measure for exploration than, say, mutual information, as in MAVEN.
Overall, there is a mixup of neighboring lines of research, terminology, and taxonomization that make the paper exceedingly difficult to follow and its contributions hard to assess. Furthermore, agent parallelization works have long shown to be faster and scalable ways to populate state-action visitation matrices, so claiming that most existing works take a single-agent perspective is a complete disregard to several rich lines of research predating this submission.

**Questions:**

How does CE compare against DIAYN?
How does CE scale with number of agent instances?
How does the KNN component scale with number of agent instances?

---

> ### Author Response · Authors · 2023-11-22
> **Reply to Reviewer ArFd [Part I]**
>
> We thank the reviewer for the constructive review and insightful comments. Responses to the questions are below.
>
> > Overall Question: "There is a mixup of neighboring lines of research, terminology, and taxonomization that make the paper exceedingly difficult to follow and its contributions hard to assess."
>
> We apologize for the inconvenience. This is indeed related to some confusions brought by our mis-usage of some terms etc. We read your comments carefully and categorize your concerns into two categories: (1) presentation related: Q1-Q3; (2) comparison related: Q4-Q6.
> We have addressed each category from several aspects as detailed below and hope the reviewer can reassess the paper based on our response and the revised paper.
>
> > Q1: "the multi-agent term is misleading ..."
>
> We apologize for the confusion and fully agree with you that the "multi-agent" term is indeed mis-leading. In our original submission, we were aware that there might be a confusion due to borrowing a term from another field, and intended to clarify it with a footnote in the original manuscript. However, it turns our that, as you commented, "repurposing a well-established terminology" does not seem to be a good strategy. Given the confusions it introduced, we fully agree with you on this point and have revised related terms and sections to address this.
> In term of coming up with a proper term, as you have suggested, our work is indeed aligned with the line of parallel agent, and we further incorporated collaboration among them, therefore we have changed the term "multi-agent" to "collaborative parallel agents" to reflect both points.
>
>
> > Q2: "the proposed method is not clearly distinguished from A3C, Ape-X, or IMPALA..."
>
> We have detailed the relationship of the proposed method and parallel agents work [A3C, Apex-X, IMPALA etc] in the respose to Common Question-1.
> In short, our work is related to parallel agent in the sense it also has parallel agents. However, the key difference is that we intent to incorporate explicit collaboration among agent and further investigated the question of how and when to collaborate.
> In more detail, our work differs from these related work in the following aspects:
> - Our work concentrates on the problem of how to collaborate between agents to improve exploration, while A3C, Ape-X and IMPALA mainly focus on improving data throughput in the parallel agent setting.
> - There is no explicit collaboration mechanism in A3C, Ape-X and IMPALA apart from the implicit way via data sharing. In our work, we attempt to answer the question of how to collaborate more effectively beyond data sharing and incorporate explicit collaboration.
> We rewrote the related sections to distinguish our work and these parallel agent work more clearly.
>
> > Q3: "claiming that most existing works take a single-agent perspective is a complete disregard to several rich lines of research predating this submission"
>
> We apologize for the confusion. We have used an imprecise term here.
> By "most existing works take a single-agent perspective", we intended to mean that that many existing work do not take an explicit collaborative perspective, either because there is only one agent (where there is no structural support for collaboration), or parallel agents are used without explicit collaborations among them.
> We have revised this sentence to make it more precise and rewrote the related sections to clarify the connections and differences of our work with previous works on parallel agents, i.e., Section 2 in the revised paper.
>
> > Q4: "underperforms Replica in HalfCheetah-V3 in terms of the variance proxy"
>
> We are sorry that we did not provide further explanation in the original paper due to the space limit, and we rewrote this section in the updated manuscript.
> Similar to the cheetah-run task in DMC, the HalfCheetah-V3 is a more like an easy task, where a single agent already achieves near-optimal performance. We added the evaluation rewards for each agent in the Table 3. We can observe that Replica agent achieves 11552.8 and CE agent achieves 11545.1, which are both very close to the optimal results in this task. Therefore, in the HalfCheetah-v3 task, the learned policies usually converge to the near-optimal policies which are less diverse.
> We followed the reviewers' suggestions and tried some other metrics, i.e., mutual information $I(S;A)$ and KL divergence $D_{KL}(\pi_i, \pi_j)$ to approximate policy diversity in the updated paper (Table 3).

---

> ### Author Response · Authors · 2023-11-22
> **Reply to Reviewer ArFd [Part II]**
>
> > Q5: "Missing baseline for distributed RL"
>
> Thanks for the suggestion. We conducted additional experiments comparing our approach to A2C in the updated Table 1. We adopt the parameter settings from https://github.com/ikostrikov/pytorch-a2c-ppo-acktr-gail/tree/master, where we run the A2C agent for 1e7 environmental steps. Notably, A2C usually underperforms other baselines, even with 10 times total environmental steps. This aligns with expectations, given that A2C-like agents (A2C/A3C/PPO) typically exhibit low sample efficiency compared with other off-policy algorithms in popular continuous control benchmarks. Our empirical results align with external evidence with on A2C and PPO, e.g.:
> - A2C is reported to have lower sample efficiency (compared to e.g. SAC) reported by StableBaselines on MuJoCo: https://github.com/DLR-RM/stable-baselines3/issues/48
> - PPO is reported to have similar performances on the DMC benchmark: https://github.com/RyanNavillus/PPO-v3
>
> > Q6: "compare to DIAYN"
>
> Thanks for the suggestion. We are sorry that we missed the information theory-based baseline, DIAYN, in the original paper. In the previous experiments, we mainly focused on methods that do not change the model architecture of the backbone algorithm, so we overlooked this important baseline which learns a skill conditioned policy $\pi(a \vert s, z)$.
> Notably, the experiment setting for DIAYN is different from the other baselines, which first pre-trains a set of skills with the pseudo-reward $r_z(s, a) = \log q_\phi(z \vert s) - \log p(z)$ and then fine-tunes the best-performing skill with the task reward. It's a little bit tricky to make fair comparison for DIAYN with other baselines if we use the same number of total environment interactions. In this supplementary experiment, we ignore this constraint and simply let the DIAYN agent run 2 times of the total environmental steps. We first use 1e6 environmental steps for pre-training and use another 1e6 environmental steps for fine-tuning. We run the experiment using the official code (https://github.com/haarnoja/sac/blob/master/DIAYN.md).
> We added the results of DIAYN to Table 1 and Figure 11. From the results, CE outperforms DIAYN in most tasks. DIAYN performs worse than CE mainly because the skills discovered in the unsupervised skill learning stage are not necessarily well aligned with the down stream tasks.
>
>
> > Q7: "how does the method scale for more agents"
>
> Thanks for the suggestion. We added results of using 16/32/64 agents in Table 7 in Appendix D.3. The performance of scaling to more agents actually depends on the tasks.
> It's notable that there is a trade-off between policy diversity and training stability:
> - With more agents, we have more diverse policies and help to solve tasks where exploration is particularly difficult, i.e., the goal-reaching fish-swim task.
> - On the other hand, the off-policyness of the sampled data increases as we have more agents. Given a sampled batch, there are only on average 1/N samples that are collected by each agent. As the number N increase, the problem becomes closer to an offline RL setting, where the optimization becomes more challenging, i.e., the performance variances increase significantly for larger N.
> Therefore, using a medium size agent set strikes a good balance of exploration and optimization.
>
>
> > Q8: "scale for KNN components wrt #agents"
> - Denote the memory buffer size as M, embedding dimension as d, and the agent number as N.
> - Then, the time complexity to compute the KNN intrinsic reward is $O(KNMd)$, which is linear w.r.t. the agent number.
> - In practice, as discussed in the response to Q7, $N$ does not need to be a large number to strike the balance between complexity and benefits brought to exploration.

---

### Author Response · Authors · 2023-11-22
**Reply to Common Questions**

> Q1: Connections and differences with respect to parallel agents, MARL and concurrent exploration.

- Thanks for the reviewers for raising this point.
  In the original submission, we deferred some of the discussions on related work to the appendix due to the space limitation. Now we revised the paper and rewrote the related work section to make the connections with parallel agents, MARL and concurrent exploration clear (Section 2 of the revised paper).
  - parallel agents [A3C, Ape-X, IMPALA]: there is no explicit collaboration among the agents. Our work is different from this body of work in that explicit collaboration is incorporated among different agents.
  - Multi-agent RL [MARL]: a typical setting is multiple agents residing in the same environment, where different agents attempt to solve a common task via cooperation, i.e., playing Dota games. In contrast to MARL, in our setting, each agent reside in its own environment. Therefore, from the perspective of settings, ours is closer to that of the parallel agents, instead of MARL.
  - concurrent exploration: our work is aligned with this category in terms of settings,  and along this line we further investigated when and how to collaborate among the different agents.
  Therefore, in summary, the proposed approach introduces explicit collaboration among parallel agents and also investigated when and how to collaborate.

> Q2: Computation complexity

- The second common concern is that the computation complexity of the proposed method increases when we use more agents in the agent set.
  - We admit that the increasing computation cost when we use a large number of agents is a weakness of the proposed method.
  - In addition, the main focus of this work is to illustrate the effectiveness of leveraging the collaborative information among different agents to improve the exploration, and we actually do not need a large number of agents.
  - Moreover, there is a trade-off between policy diversities and training stability when we increase the agent number. We provide more detailed discussion about this point in Appendix D.3.
  - To mitigate the computation complexity issue, we adopt some implementation tricks, including vectorized models with JIT compilation, to reduce the wall-clock running time.

---

### Author Response · Authors · 2023-11-22
**General reply**

We would like to thank all the reviewers for their time and efforts on providing us with constructive feedback to improve this paper. We will first provide a response summary here and later address each reviewer's concerns in separate responses. In the following replies, we will use R1/R2/R3/R4 to refer to the reviewer ArFd, JJ8w, AU7q and 6JRU, respectively.

We are happy to see that reviewers generally agree that:
(1) this work studies an interesting / important topic [R2, R4];
(2) the presentation is clear [R2, R3, R4]; and
(3) the experiment results are promising [R1, R2, R3, R4].

According to the constructive comments from the reviewers, we have:
1. expanded and rewrote the related work to clarify the relationship of our work with others [R1, R2, R3, R4];
2. added new experiments to use mutual information and KL divergence as metrics to approximate policy diversity [R1, R4];
3. added an information theory-based baseline, DIAYN [R1];
4. added a distributed RL baseline, A2C [R1];
5. added new experiments for larger agent numbers [R1];
6. added new experiments with a multi-head variant with parameter sharing [R1, R4];
7. added new experiments for RND in Figure 4 [R2];
8. added new experiments on the Atari games [R2];
9. plot more trajectories in Figure 5 [R2];
10. added missing references on the concurrent exploration and parameter-sharing based MARL [R2, R3].

---

### Meta-Review · Area_Chair_dvVV · 2023-12-08

**Metareview:**

The paper introduces a collaborative exploration framework in reinforcement learning, using multiple agents to enhance exploration. Key features include a collaborative reward generator that encourages agents to explore novel states, leading to collaboration and specialization. The framework enables shared information among agents and coordinated data collection. Experimental results on DeepMind Control Suite tasks demonstrate superior performance compared to single-agent baselines and other intrinsic reward methods.

The paper innovatively tackles the exploration problem in single-agent reinforcement learning by adopting a collaborative perspective. Its algorithm stands out for its simplicity and clarity, backed by strong empirical results showcasing its effectiveness over baselines like RND and ICM. The flexibility of the framework and its adaptability to different intrinsic reward designs are notable strengths. Evaluations on established DMC benchmark tasks enhance the study's credibility, and the collaborative rewards mechanism contributes to inducing specialization between agents. The general applicability of the collaborative idea to diverse reinforcement learning algorithms is recognized as a valuable contribution.

On the other hand, the paper shows significant weaknesses that necessitate attention. The choice of the term "multi-agent" lacks strong justification, departing from established usage. Issues in the related works section include omissions of key references and a notable absence of MARL algorithms in the evaluation. Furthermore, crucial aspects such as computational complexity and the observed increase in computation cost are not sufficiently addressed. An extension of the related work section is suggested to provide a more comprehensive overview. Addressing these concerns is pivotal for enhancing the paper's conceptual clarity and integration into the broader field of reinforcement learning research.

The authors are encouraged to follow reviewers' suggestions while preparing a new version of their paper.

**Justification For Why Not Higher Score:**

The reviewers share some concerns about the positioning of this work with respect to the state of the art.

**Justification For Why Not Lower Score:**

N/A

---

### Decision · Program_Chairs · 2024-01-16

Reject